# Latent Profile Analysis of Mental Health among Chinese University Students: Evidence for the Dual-Factor Model

**DOI:** 10.3390/healthcare11202719

**Published:** 2023-10-12

**Authors:** Yumei Jiang, Chen Ding, Bo Shen

**Affiliations:** 1School of Physical Education and Sports, Central China Normal University, Wuhan 430079, China; jiangyumei@hust.edu.cn (Y.J.); boshen@mail.ccnu.edu.cn (B.S.); 2School of Physical Education, Huazhong University of Science and Technology, Wuhan 430074, China; 3College of Education, Wayne State University, 42 W. Warren Ave., Detroit, MI 48202, USA

**Keywords:** dual-factor model, mental health, Chinese university students, academic emotions

## Abstract

The dual-factor model of mental health has garnered substantial support, positing the necessity of encompassing both negative (e.g., psychological problems) and positive (e.g., well-being) indicators in comprehensive evaluations of people’s mental health. Nonetheless, the nature of the profiles and predictors (such as academic emotions) during four years of university life lack clarity, hampering a profound understanding of mental well-being among university students. This research included 135 items designed to assess an array of depression symptoms, negative emotional experiences, life satisfaction, positive emotional experiences, and academic emotions. First, this research affirmed the applicability of the dual-factor model in the context of Chinese university students (N = 2277) with the utilization of confirmatory factor analysis (CFA). Furthermore, latent profile analysis (LPA) was employed to delineate prevalent constellations of psychological symptoms and subjective well-being within participants. The outcomes unveiled the existence of three distinct clusters: (1) Complete Mental Health, (2) Vulnerable, and (3) Troubled. Third, by employing R3stept on academic emotions and mental health classifications, this study revealed that there were associations between this variable and specific amalgams of psychological symptoms and well-being levels. These findings bear influence on the practice of mental health screening and the identification of individuals necessitating targeted interventions.

## 1. Introduction

In the current phase, university students find themselves within an environment marked by social transitions, economic reforms, and intensified competition. This context coincides with a crucial life stage characterized by rapid intellectual, vocational, and psychosocial development, rendering students highly susceptible to psychological challenges such as anxiety, low self-esteem, and depression [1]. Mental health issues among this demographic not only lead to academic disengagement, leave of absence, and even dropout but also exert a substantial and undeniable impact on both their scholastic pursuits and overall well-being [2,3,4,5]. Concurrently, academic emotions, as the emotional experience intertwined with students’ learning endeavors, stand as a pivotal component of their daily emotional landscape [6]. Academic emotions substantially influence students’ academic performance and personality development, and can even reverberate throughout their lifetime well-being [7,8]. To adequately support young individuals through this critical developmental stage, it becomes imperative to comprehend the following: (1) Who is particularly susceptible to compromised mental health? and (2) What is the factor (particularly school-related predictor) that determines group disparities in mental health profile?

Extensive research consistently underscores that the mental health status of Chinese university students is far from sanguine, with a discernible upward trend in the commonness of mental disorders among this demographic [9,10]. However, within this population, a substantial degree of heterogeneity prevails [11]. Traditional empirical inquiries into mental health predominantly rely on psychopathological indicators. This diagnostic system fails to ensure a holistic understanding of mental health, due to the absence of positive indicators (such as well-being) [12,13]. As the positive psychology movement gains momentum, this approach, which fixates on measuring psychopathological symptoms or psychological dysfunctions, faces increasing scrutiny. Positive psychology seeks to elucidate a spectrum of factors that foster psychological well-being, such as subjective well-being, optimism, life satisfaction, and self-efficacy, advocating for their inclusion in the evaluation of mental health to foster a more integrated and accurate understanding [12]. Building on this foundation, international scholars have proposed the dual-factor model of mental health (DFM) [14,15,16,17]. The DFM postulates that mental health comprises two distinct but intertwined dimensions: psychopathology and well-being [14]. According to this model, researchers should concurrently examine both negative and positive indicators.

Currently, research focusing on empirical interventions on the basis of the dual-factor model of mental health remains scarce, especially concerning the unique and expanding demographic of Chinese university students [18]. Thus, it remains a pertinent query whether the dual-factor model of mental health, predicated on the symptoms of psychopathology and well-being, is suitable for gauging the mental health of students in Chinese universities. In other words, does this emerging theoretical model present advantages over traditional assessment paradigms? Moreover, what is the nature of the profile of Chinese university students’ mental health? And does the differences in academic emotions predict the mental health statuses of Chinese university students? These questions warrant further exploration and inquiry.

### 1.1. The Dual-Factor Model of Mental Health

The dual-factor model of mental health employs psychopathological symptoms as negative psychological indicators and subjective well-being as positive psychological indicators. By doing this, the dual-factor model addresses the overreliance on singular negative indicators within a traditional psychological health diagnosis framework. This approach provides a more accurate and comprehensive understanding of an individual’s mental health status [14]. Psychopathological indicators encompass internalizing issues (such as depression and anxiety) and externalizing problems (such as behavioral disorders), while subjective well-being encompasses positive emotions and life satisfaction [19]. Previous studies based on the dual-factor mental health framework have classified the population into four types: (1) Complete Mental Health, individuals who exhibit low psychopathological symptoms and high subjective well-being, reflecting strong psychological and social functioning; (2) Vulnerable, individuals who display low psychopathological symptoms and also low subjective well-being. This group does not meet the diagnostic criteria of psychopathology but remains at risk of future psychological issues, and is often overlooked by traditional psychological health models; (3) Symptomatic but Content, individuals who exhibit high psychopathological symptoms and high subjective well-being, indicating a capacity for psychological self-repair despite the presence of certain psychological symptoms; and (4) Troubled, individuals who demonstrate psychological symptoms such as anxiety and depression [20].

Research has substantiated the advantages of the dual-factor approach to mental health over conventional approaches focused solely on problems. Notably, studies have demonstrated that “Complete Mental Health” individuals experience more relaxation-oriented academic emotions and fewer feelings of anxiety and helplessness than students in the other three groups [3]. Additionally, scholars have identified an often-overlooked cohort of adolescents (with low subjective well-being and low psychopathology) who are, nevertheless, prone to academic and behavioral issues, performing no better than the most troubled adolescent group (e.g., Antaramian et al., 2010) [21]. These findings underscore the notion that psychopathology alone is inadequate for maximizing functioning. The recognition of the “Vulnerable” group is especially significant, as traditional assessment systems based on psychopathology measures could disregard these individuals, losing the chance for interventions before students’ psychological disturbance is severe enough to require clinical treatment. Furthermore, research has revealed that “Symptomatic but Content” individuals can significantly reduce negative emotions through aerobic exercise interventions. Moreover, those who did not undergo exercise intervention still experienced a significant reduction in negative emotions. Psychologists such as Keyes et al. (2005) [16] suggested that, although “Symptomatic but Content” individuals may possess certain psychological ailments, their relatively high levels of positive forces like happiness might facilitate spontaneous recovery over time. Various researchers have concluded that the examination of an individual’s mental health within the dual-factor model can provide a more nuanced and comprehensive depiction of their mental health functioning [22].

### 1.2. Methodology of Dual-Factor Mental Health Investigating

To explore dual-factor mental health, researchers have conventionally identified the aforementioned subgroups using sample or norm-based cut-off points [23]. Researchers (e.g., Ai Chunyan, 2021; Suldo and Shaffer, 2008; and Guo Kangwei, 2017) categorized individuals by devising a logically derived dichotomous classification scheme for both psychopathological and subjective well-being dimensions [11,15,24]. Diverse criteria have been employed for classification.

For instance, Antaramian et al. (2010) [21] adopted established decision thresholds for norm-referenced measures. They classified participants as part of the “Complete Mental Health” group if they had a T score of 60 or lower on psychopathological symptoms scales and a T score of 40 or higher on well-being scales; the “Vulnerable” group if they had a T score of 60 or lower on psychopathological symptoms scales and a T score of 40 or lower on well-being scales; the “Symptomatic but Content” group if they had a T score of 60 or higher on psychopathological symptoms scales and a T score of 40 or higher on well-being scales; and the “Troubled” group if they had a T score of 60 or higher on psychopathological symptoms scales and a T score of 40 or lower on well-being scales. Wang et al. (2016) [25] used values derived from individuals’ sample means. They classified participants into different groups based on the mean scores of the combined anxiety and depression scales (mean = 1.81) and the mean scores of the combined happiness and life satisfaction scales (mean = 27.03). However, the cut-off points approach mentioned above which was based on the sample or norm is not without its limitations. Imposing cut-off points can lead to minimal differentiation: individuals with comparable scores on either side of the cut-off might be assigned to separate groups [20,26]. Additionally, these approaches pre-assume the nature of the subgroups to be identified based on theory instead of data [27].

Recently, methodologies like latent class analysis (LCA) and latent profile analysis (LPA) have emerged to identify subgroups of children and adolescents exhibiting similar patterns of dual-factor mental health (such as Petersen et al., 2022 [20]; Petersen et al., 2020 [26]; and Berlin et al., 2013 [28]). These approaches present several benefits over cut-off point techniques. Primarily, LCA and LPA are model-driven clustering techniques that uncover underlying subgroups based on diverse variables of the indicator. They uncover subgroups that are internally homogeneous and externally heterogeneous [29].

Presently, the utilization of LCA and LPA within the context of the dual-factor model is primarily documented within Western contexts, with limited references within Chinese literature. One exception is that Zhou et al., (2020) [29], using LPA, conducted a survey encompassing 1009 middle school students in China. The outcomes delineated three distinct groups: Flourishing Youth (characterized by low depressive and anxiety symptoms, and high self-esteem and life satisfaction), Vulnerable Youth (displaying low depressive and anxiety symptoms, and low self-esteem and life satisfaction), and Troubled Youth (marked by high depressive and anxiety symptoms, and low self-esteem and life satisfaction). Therefore, it is essential to further explore the applicability of this classification approach to Chinese university students and to ascertain its potential implications for measuring and intervening in the mental health of our country’s university students.

### 1.3. Academic Emotions and the Psychological State of University Students

Academic emotions hold significant implications for the psychological well-being of students as an integral facet of their daily emotional experiences. Positive academic emotions are deemed a prerequisite for normal adolescent learning and contributes to the cultivation of a robust personality [4]. In 2002, German psychologist Pekrun et al. [30] first coined the term “academic emotions” to encompass emotions directly linked to academic learning, classroom instruction, and academic achievement. Pekrun et al.’s (2002) research identified nine distinct emotions constituting academic emotion, further categorized into four groups based on valence and arousal: positive high-arousal, positive low-arousal, negative high-arousal, and negative low-arousal.

Academic emotions have been investigated in conjunction with mental resilience among young individuals. Notably, positive correlations have been found between positive academic emotions and mental resilience, and negative correlations between negative academic emotion and mental resilience. Furthermore, academic emotions have been found to predict the mental resilience and psychological well-being of secondary school students (e.g., Jiang and Xu (2017) [31]). While many studies have focused on the relationship between academic emotions and positive psychological states, only a limited number have explored the interplay between academic emotions and the dual-factor mental health model, which concurrently examines symptoms of distress and subjective well-being. For instance, research indicates that students with different mental health statuses exhibit significant differences in terms of perceived academic stress and academic emotions [6]. “Complete Mental Health” individuals experience lower levels of academic stress compared to “Vulnerable” individuals. Additionally, “Complete Mental Health” individuals experience lower levels of academic stress compared to “Symptomatic but Content” individuals. Moreover, across all five dimensions of academic stress, “Complete Mental Health” individuals report lower levels than “Troubled” individuals. Comparative results regarding academic emotions reveal that “Complete Mental Health” individuals experience a greater sense of relaxation in their academic emotions and fewer feelings of anxiety and helplessness than students in the other three groups [6]. Furthermore, a four-month longitudinal study found that the initial anxiety and helplessness experienced in academic emotions by first-year students can predict their end-of-term psychological well-being. Conversely, the initial psychological well-being level predicts the subsequent experience of relaxed and helpless academic emotions at the end of the term [32]. This underscores the reciprocal influence between academic emotions and students’ psychological well-being, with academic emotion serving as a reliable predictor of their psychological state.

### 1.4. The Present Study

The present study aims to expand upon prior research by offering additional evidence regarding the applicability of the dual-factor mental health model in the context of Chinese university students. This study thus addressed three major aims: First, to verify the applicability of the dual-factor model among Chinese college students, the current study used confirmatory factor analysis (CFA) to investigate which goodness-of-fit statistics were superior between the single-factor model and dual-factor model of mental health. Second, instead of using an a priori cut-point classification method as in previous studies, this study aimed to examine Chinese college students’ reports of depression symptoms, negative emotional experiences, life satisfaction, and positive emotional experiences within the dual-factor mental health framework by using latent profile analysis (LPA) which is a model-based clustering technique. Third, in an effort to identify key school-related factors associated with the development of the mental health patterns among Chinese college students, this study aimed to examine the specific predictive role of academic emotions on the identified profiles by using multinomial logistic regression.

## 2. Materials and Methods

### 2.1. Participants

The study participants were drawn from a higher education institution in Wuhan, China. Employing a clustered sampling approach organized by first-year classes in the targeted college, 2850 college students in total were selected to take the questionnaire survey. 

After excluding invalid responses, a total of 2277 valid questionnaires were retained, resulting in a response rate of 79.9%. Among the participants, 1402 were male, accounting for 61.6% of the sample; 847 hailed from rural areas, representing 37.2% of the sample; and, notably, the majority, 1598 students, were enrolled in science and engineering disciplines, constituting 70.2% of the sample (considering that the institution is a science-and-technology-oriented university, this proportion is deemed reasonable) (Table 1).

### 2.2. Measures

#### 2.2.1. Satisfaction with Life Scale (SWLS)

The Satisfaction with Life Scale (SWLS), developed by Diener et al. in 1985 [33], was utilized in this study to assess overall life satisfaction and measure the components of cognition of subjective well-being. The SWLS is a unidimensional scale comprising 5 items, scored on a 7-point Likert scale (“1” indicating “strongly disagree” and “7” indicating “strongly agree”). Higher scores indicate higher levels of life satisfaction. This widely employed questionnaire demonstrates robust reliability and validity, effectively measuring life satisfaction among the general population [34]. The internal consistency of the questionnaire in the present study was assessed with a Cronbach’s coefficient alpha of 0.808.

#### 2.2.2. Positive Affect Scale (PAS)

Derived from the Positive Affect and Negative Affect Scale (PANAS) by Watson and Clark (1988) [35], the Positive Affect Scale (PAS) includes 10 items describing positive emotional adjectives. Participants were instructed to rate the extent to which they experienced these emotions during the past week using a 5-point Likert scale. Responses ranged from “1” indicating “very slightly or not at all” to “5” indicating “extremely.” The internal consistency of the questionnaire in this study yielded a Cronbach’s coefficient alpha of 0.869.

#### 2.2.3. General Health Questionnaire (GHQ-12)

The revised 12-item General Health Questionnaire (GHQ-12), developed by Goldberg (1997) [36], was employed in this study to assess participants’ recent general psychological well-being. This scale, considered one of the most widely used tools for measuring psychological issues, consists of positively and negatively worded items [37], with 6 items each, including positive items (e.g., “being able to concentrate on whatever I’m doing”) and negative items (e.g., “losing sleep due to anxiety”). Scoring followed the GHQ-12 standard method, where the first two items are scored as 0 and the latter two as 1, resulting in a total score ranging from 0 to 12. Higher scores indicate poorer psychological health. The questionnaire’s internal consistency in this study was indicated by a Cronbach’s coefficient alpha of 0.849.

#### 2.2.4. Self-Rating Depression Scale (SDS)

The Self-Rating Depression Scale (SDS) was founded by Zung of Duke University School of Medicine in 1965 and is currently one of the most widely used self-rating depression scales [38]. It is a brief self-administered survey designed to quantify an individual’s state of depression. Widely applied as a depression assessment tool, the scale comprises 20 items, with 10 items scored in the positive direction and 10 in the reverse. SDS employs a 4-level scoring system, primarily assessing the frequency of symptom occurrence, with ratings as follows: “1” for “rarely or none of the time”, “2” for “some of the time”, “3” for “a good part of the time”, and “4” for “most or all of the time”. The internal consistency of the SDS questionnaire in this study was reflected by a Cronbach’s coefficient alpha of 0.828.

#### 2.2.5. Academic Emotion Scale

The “Academic Emotion Questionnaire”, jointly authored by Dong and Yu in 2007 [6], consists of 88 items organized into four subscales: Positive High-Arousal Academic Emotion, Positive Low-Arousal Academic Emotion, Negative High-Arousal Academic Emotion, and Negative Low-Arousal Academic Emotion. These subscales provide distinct perspectives on students’ academic emotions, contributing to a comprehensive investigation. The scoring system for this questionnaire employs a 5-point scale, with the scores summed separately for positive and negative academic emotions. Higher scores in positive academic emotions reflect a more positive attitude towards academics, while higher scores in negative academic emotions suggest a more negative emotional response to academic demands. The reliability of the four academic emotion subscales in this questionnaire is indicated by the following Cronbach’s coefficient alphas: Negative High-Arousal (e.g., emotions of shame, anxiety, and anger) 0.739, Positive High-Arousal (e.g., emotions of interest, pleasure, and hope) 0.912, Negative Low-Arousal (e.g., emotions of disappointment and boredom) 0.906, and Positive Low-Arousal (e.g., emotions of pride and relaxation) 0.898.

#### 2.2.6. Methods

The data analysis encompassed three distinct stages, and all data analyses were conducted by using Mplus 8.7. Initially, confirmatory factor analysis (CFA) was conducted to assess the validity of the dual-factor model of mental health in the context of Chinese university students. The proposed parameters were compared between the dual-factor model of mental health and conventional singular-factor model. 

Subsequently, a data-driven approach, latent profile analysis, was conducted using the collected samples to ascertain the latent classes within the data. In this pursuit, fit indices were taken into consideration alongside interpretations of class structures. The objective was to identify the number of classes that produced the most meaningful, succinct, and statistically robust model, following the guidelines established by Collins and Lanza in 2010 [27]. There are four steps involved in conducting LPA: (1) Data inspection: All indicators with different scoring methods were transformed into z-scores for LPA analysis, so as to facilitate comparison and interpretation. (2) Iterative evaluation of models and model fit: Using the Satisfaction with Life Scale (SWLS), Positive Affect Scale (PAS), General Health Questionnaire (GHQ-12), and Self-Rating Depression Scale (SDS) as indicator variables, latent profile analyses (LPAs) with 1 to 5 categories were fitted to estimate and compare the psychological health characteristics of university students. Considering fit indices and category structure, the most concise, meaningful, and statistically sound category structure was selected [27]. The primary fit indices for model adequacy testing included the log-likelihood (LL) test, as well as information criteria such as the Akaike information criterion (AIC), Bayesian information criterion (BIC), sample-adjusted BIC (aBIC), and entropy index. Likelihood ratio test indices, specifically the Lo–Mendell–Rubin (LMR) and bootstrap likelihood ratio test (BLRT) statistics, were also employed. Smaller values of LL, AIC, BIC, and aBIC indicate better model fit [39]. An entropy index approaching 1.0 suggests improved predictive ability, while an entropy value near 0.8 indicates classification accuracy exceeding 90% [40]. The statistical significance (*p*-value) of LMR and BLRT suggests that a model with k classes significantly outperforms a model with k-1 classes [41]. (3) Model interpretability: The average membership probabilities for each latent class consistently exceed 0.80, indicating a high level of classification accuracy. (4) Names of patterns in a retained model: This involves identifying individual memberships and naming each category [42]. Finally, based on the previous two steps, using academic emotions as the independent variable and mental health profiles as the dependent variable, the relationship between the two variables was tested using nultinomial logistic regression [43]. This analytical step aimed to discern the connection between different academic emotional experiences and the identified latent profiles of students.

## 3. Results

### 3.1. Establishment and Examination of the Dual-Factor Model of Mental Health

To evaluate the rationality of the dual-factor model and validate its superior measurement of mental health compared to the traditional model, this study followed the model construction strategy proposed by Jöreskog and Sörbom [44] to validate various structural models. Based on the foundation of the dual-factor model of mental health and drawing inspiration from prior research [3], two structural models were constructed.

Model 1 represented the single-factor model of mental health. In this model, a latent variable representing overall mental health was created. Positive aspects of life satisfaction (SWLS) and positive emotions (PAS) were positively loaded onto this latent variable, while negative aspects of depressive symptoms (SDS) and psychological distress symptoms (GHQ-12) were negatively loaded, as illustrated in Figure 1 and Figure 2.

Model 2 represents the dual-factor model of mental health, encompassing two latent variables: Positive Psychological Well-Being and Negative Psychological Well-Being. In this model, the observed variable corresponding to satisfaction with life is the score from the Satisfaction with Life Scale (SWLS) for the respondents, while the observed variable corresponding to positive emotions is the score from the Positive Affect Scale (PAS). On the other hand, the observed variable corresponding to negative psychological issues is the score from the General Health Questionnaire (GHQ-12) for the respondents, and the observed variable corresponding to depressive symptoms is the score from the Self-Rating Depression Scale (SDS). Both positive and negative psychological well-being factors collectively influence overall psychological well-being, and there exists a mutual influence between these two factors.

To verify the rationality of the dual-factor model of mental health, this study conducted a comparison between Model 1 and Model 2. The overall model fit was determined by various goodness-of-fit statistics, as well as its chi-square value. When the sample size is substantial, a chi-square test (χ^2^/df) below 5 indicates a good fit of the data. The criteria for each of the goodness-of-fit indices are as follows: (1) CFI values should not be less than 0.95 for an acceptable fit; (3) NFI should exceed 0.95, indicating a good fit; (4) RMSEA should be as small as possible, and any value greater than 0.08 should be interpreted as an unreasonable fit; and (5) SRMR should not be higher than 0.08, suggesting an acceptable fit [45,46,47,48]. 

Using confirmatory factor analysis (CFA) in Mplus 8.7, validation was conducted for the different models. The outcomes are presented in Table 2. The results demonstrate that the fit indices of the dual-factor model of mental health surpass those of the single-factor model. 

### 3.2. Classification and Statistics of Student Psychological Well-Being States

#### 3.2.1. Data Inspection

Before conducting the data analysis, all indicator scores were standardized, in order to reduce the impact of the dimensional inconsistency of the selected instruments (such as the Satisfaction with Life Scale (SWLS), Positive Affect Scale (PAS), General Health Questionnaire (GHQ-12), and Self-Rating Depression Scale (SDS)) on theLPA results.

#### 3.2.2. Iterative Evaluation of Models and Model Fit

Table 3 presents the goodness-of-fit measures utilized by the authors to determine the optimal number of classifications for our dataset. An examination of these results revealed that a three-class solution was deemed the most suitable for our data. Commencing with a one-class model (C = 1), the authors progressively increased the number of potential classes (C = 2, C = 3, C = 4, C = 5), simultaneously fitting these potential class models separately. Notably, the values of AIC, BIC, and aBIC consistently decreased with an increasing number of classes. However, the rate of decrease plateaued beyond three classes, signifying that the three-class model exhibited the best fit. Additionally, the entropy index remained above 0.80 for class counts ranging from two to five class models. The LMR and BLRT statistics for the three-class model were also statistically significant. Regarding class probabilities, when dividing into four or five classes, the class proportions fell below 5%, indicating insufficient representativeness. Based on these findings, selecting three classes as the number of latent categories was deemed appropriate. Moreover, in the three-class model, the three groups constituted 47.6%, 44.1%, and 8.3% of the sample, respectively.

#### 3.2.3. Model Interpretability

As shown below (Table 4), the average membership probabilities for each latent class consistently exceed 0.80, indicating a high level of classification accuracy.

#### 3.2.4. Names of Patterns in a Retained Model

Figure 3 presents the estimated mean plots for the three-class latent profile analysis solution for our data. Based on the pattern of mean scores across depression symptoms (SDS), negative emotional experiences (GHQ-12), life satisfaction (SWLS), and positive emotional experiences (PAS), the following labels were offered for the three emerging classes: (1) the “Vulnerable” group: This group exhibits low levels of depressive symptoms and passive emotional experiences, as well as low life satisfaction and positive emotional experiences; (2) the “Flourishing” group: This group demonstrates low levels of depressive symptoms and passive emotional experiences, coupled with high life satisfaction and positive emotional experiences; and (3) the “Troubled” group. Members of this group display high levels of depressive symptoms and passive emotional experiences, alongside low life satisfaction and positive emotional experiences. 

### 3.3. Predictive Role of Academic Emotions

Table 5 includes odds ratios (ORs) for the predictor of academic emotions. When comparing “Flourishing” to “Vulnerable” individuals, for every one-unit increase in negative high-arousal scores, the probability of being categorized as “Flourishing” decreases by 5.1%. Conversely, for each one unit increase in positive high-arousal and positive low-arousal scores, the likelihood of belonging to the “Flourishing” category increases by 10.1%. In contrast, when “Troubled” individuals are compared to “Vulnerable” individuals, for every one-unit increase in negative high-arousal scores, the probability of falling into the “Troubled” category increases by 8.2%. However, for each one-unit increase in positive high-arousal scores, the probability of being categorized as “Troubled” decreases by 8.3%. Similarly, with each one-unit increase in positive low-arousal scores, the likelihood of being classified as “Troubled” also decreases by 7.8%. Notably, negative low-arousal scores do not exhibit predictive effects. 

These findings underscore that positive high-arousal and positive low-arousal emotional experiences in the academic context have a favorable predictive role in psychological well-being. Conversely, negative high-arousal academic emotional experiences have an adverse predictive effect on psychological well-being. Individuals who report more positive academic emotions such as pride, happiness, hope, and relaxation tend to have higher levels of psychological well-being. On the other hand, those who experience negative academic emotions like anxiety, depression, fatigue, and helplessness often encounter various psychological challenges. 

## 4. Discussion

Physiologically, college students are considered adults; however, due to their prolonged role as students, they have been sheltered by parents and teachers, resulting in limited social exposure and experiences. On the contrary, they represent a demographic with relatively higher cultural levels within society, displaying active thinking and setting high expectations for themselves. When their lofty ideals encounter the stark realities, various psychological confusions, conflicts, and contradictions emerge, disrupting their psychological equilibrium and posing threats to their mental well-being. Consequently, the significance of psychological health education for college students is paramount [49]. 

The mental health dual-factor model has underscored the importance of encompassing both negative and positive mental health indicators to comprehensively grasp the mental well-being of adolescents. However, due to certain methodological limitations (such as the classification approach derived by a priori logic) existing in previous studies, the underlying nature of profiles related to dual-factor mental health among Chinese university students remains unclear. Furthermore, limited knowledge exists concerning the predictors, particularly those related to academic life, that contribute to the resulting profiles in dual-factor mental health. Gaining a deeper insight into these dimensions would provide a distinctive perspective for evaluating the mental health statuses of university students comprehensively, thereby aiding in the development of effective upstream prevention strategies.

### 4.1. Advantages of the Dual-Factor Model of Mental Health

The present study validated different models of mental health and found that the dual-factor model exhibited satisfactory goodness-of-fit statistics. The results lend support to the theoretical foundation of the dual-factor model, which posits psychological dysfunction and subjective well-being as two crucial indicators of mental health that are intertwined and indispensable. This adds further substantiation to the applicability of the dual-factor model among Chinese university students. 

The dual-factor model of mental health has provided a more intricate and comprehensive depiction of mental health functioning compared to conventional approaches. Within this study, three distinct clusters were identified based on varying degrees of psychological symptoms and subjective well-being. The “Flourishing” group was considered as the most favorable mental health profile, as it displayed high levels of subjective well-being along with minimal psychopathological symptoms. Conversely, participants characterized by low subjective well-being and heightened psychopathological symptoms were designated as the “Troubled” group. These two groups align with the categorizations typically derived from a conventional single-factor model of mental health. However, this study revealed an additional group, reinforcing the dual-factor model’s validity by indicating that positive well-being and psychopathological symptoms are not polar opposites on a singular continuum. Labeled as “Vulnerable”, this group exhibited insignificant psychopathological symptoms, yet their subjective well-being was notably low. Utilizing a traditional approach, these individuals might be categorized as mentally healthy; however, their low subjective well-being sets them apart from their “Flourishing” counterparts. This underscores that the absence of psychopathological symptom alone does not equate to optimal mental health functioning.

In line with one of the researches (e.g., Antaramian et al., 2010) [21], the proportion of individuals within this group surpasses other groups that are found in studies, hinting that college students might be particularly susceptible to a lack of positive well-being, even without apparent psychopathological symptoms. 

### 4.2. Profiles of Dual-Factor Model of Mental Health among Students of Chinese University

The present study employed latent profile analysis (LAP) to ascertain dual-factor mental health profiles among Chinese university students. Substantial variability was identified for the four mental health levels (e.g., SWLS, PAS, SDS, and GHQ-12), leading to the identification of three distinct groups: Flourishing, Vulnerable, and Troubled. While this study supported a multidimensional mental health model, it diverged from the anticipated four-group solution. The emergence of a three-group classification was not consistent with previous research in both Western and Eastern contexts, such as the four-group classification, including “Flourishing”, “Troubled”, “Vulnerable”, and “Symptomatic but content”, proposed by Antaramian et al. (2010) [21] and Xiong et al. (2017) [50]. It is important to note, however, that previous works of research that have found four mental health groups are usually based on the use of pre-determined, clinical cut-off points to generate mental health profiles among participants.

Interestingly, the outcome of a three-group classification echoed the findings of Zhou et al. (2020) [29], which also utilized latent profile analysis (LAP) as a more sophisticated and accurate classification approach. Similar to our study, their research also did not yield a class of students categorized as “Symptomatic but Content” (i.e., individuals with high well-being and high psychopathological symptoms). Zhou et al. (2020) offered two potential explanations for this deviation. One interpretation points to cultural disparities arising from the collectivistic orientation of Chinese students. Such cultural variances between individuals from individualistic and collectivistic cultures might lead to differences in the nature and consequences of subjective well-being constructs [51]. Alternatively, methodological differences could play a role, particularly differences in the choice of indicators representing negative mental health across studies. Both our study and Zhou et al.’s (2020) [29] study only include internalizing behaviors (such as depression symptoms) as the negative indicators, while most studies include both measures of internalizing and externalizing behaviors (such as conduct problems and attention deficit). This omission of externalizing measures might have obscured the identification of a “Symptomatic but Content” group. Consequently, future investigations should encompass indicators of externalizing behaviors among negative mental health measures to clarify the outcomes of this study.

In summary, considering the findings of both this study and Zhou et al. (2020) [29], wherein the conventional four dual-factor mental health groups were not replicated, it is prudent not to automatically categorize students into these four predefined groups during mental health assessments. Instead, educational professionals should adopt more sophisticated mental health screening systems (such as model-driven approaches) to identify specific student groups within distinct settings that require targeted assistance.

### 4.3. The Effects of Academic Emotion in School

This study delved into the ramifications of university students’ encountered academic emotions within the school environment to shed light on the origins of the identified groups and to provide insights for refining assessment and intervention strategies. Consistent with prior research (e.g., Zhou et al., 2020; Chen et al., 2014; and Bakker et al., 2010) [29,52,53], the findings underscored that heightened levels of positive high arousal and positive low arousal experienced within the academic setting acted as protective factors, fostering a positive mental health status among students. Conversely, elevated levels of academic stress, represented by negative high-arousal emotions, are risk factors, contributing to a negative mental health status among students. In essence, the academic emotions experienced by Chinese university students within the school context hold the potential to serve as potent predictive indicators for the development of their dual-factor mental health.

### 4.4. Strength, Limitations, and Further Research

The current study exhibited three significant strengths in comparison to previous research. The first strength lies in the substantial sample size of Chinese college students. The second major strength was the adoption of an individual-centered approach to explore the psychological well-being and the role of academic emotions among college students. However, this study does possess certain limitations and suggests directions for future research. Firstly, all data relied on self-reporting by students, which might have implications for internal validity. Future studies could collect data from multiple sources, such as teachers and parents, to enhance measurement validity by mitigating common method biases and the potential impact of overly positive self-statements [54]. Secondly, while this study incorporated four prevalent and significant mental health indicators (namely, symptoms of depression, negative emotional experiences, life satisfaction, and positive emotional experiences), it is important to acknowledge that externalizing behaviors were not measured. Future research could consider including these additional indicators to provide a more comprehensive understanding of college students’ mental health statuses. Lastly, all participants were from the same grade of college. Future research could focus on the differences in mental health status among different grades in the analysis of more samples. Moreover, this study was based on the dual-factor model of psychological well-being for screening mental health. Future research could combine this with targeted intervention techniques to comprehensively validate the feasibility and practicality of the dual-factor model of psychological well-being.

### 4.5. Implication

The dual-factor model (DFM) of mental health emphasizes that it is not sufficient to judge all individuals within a group based solely on indicators of pathological or positive psychological tendencies. Therefore, within the mental health education system of universities under the framework of the DFM, changes are required in crisis prevention and intervention strategies compared to previous approaches:

(1) Strengthening the comprehensive tracking and evaluation of student mental health based on the dual-factor model of mental health.

The results of this study indicate that colleges should focus on both the negative and positive aspects of college students, comprehensively tracking and evaluating their mental health status. The screening scales selected in the annual psychological census work in colleges need to change. In the past, the practice in the psychological census of the college mental health education system under the traditional mental health model was to directly select some scales such as SCL-90, UPI, SDS, and SAS that tend to judge diseases clinically to screen students’ psychological problems. Populations with low depression symptoms and low happiness are not easily detected in traditional psychological measurements. Colleges should actively try to add some scales that reflect positive emotions, such as subjective well-being scales, positive and negative emotion scales, social support scales, etc., during the psychological census. This facilitates the screening of different categories of individual mental health under the dual-factor model of mental health. 

Moreover, colleges should establish personal psychological files, carry out various forms of education and counseling according to the different needs of different students, and reduce their psychological problems while cultivating positive psychological qualities. 

(2) Pay attention to different types of college student mental health groups for precise intervention.

The results of the current study showed that there is a large heterogeneity within the college student group. Therefore, colleges should use different structured interviews and psychological counseling for individuals with different mental health statuses screened out by the DFM’s psychological census method, making interviews or psychological counseling for individuals in various subtypes more targeted. For example, the completely pathological individuals (the “Troubled” group) not only have high mental symptoms but also low life satisfaction, poor adaptability, and a lack of ability to cope with greater challenges. We should apply maximum-intensity psychological intervention measures to this group of students, cultivate their problem-solving ability and pro-social behavior, enable them to effectively manage their emotions, and reduce the depression level of disease groups, thereby protecting them from adversity. While the “Vulnerable” group may not have high mental symptoms, they show very low life satisfaction and may be affected by negative information. Notably, this group was the biggest group among participants, constituting 47.6% of the sample. In the process of psychological counseling, we should pay attention to the cultivation of positive forces instead of just focusing on eliminating negative symptoms. 

(3) Focus on college students’ academic emotions and build harmonious teacher–student relationships. 

The present findings indicate that academic emotions experienced in school serve as strong predictors of the dual-factor mental health profiles among Chinese college students. These results suggest that educational professionals in China could focus on fostering positive academic emotions and mitigating academic stress within the school environment. Teachers, in particular, can play a crucial role by effectively steering students towards positive academic attributions, which are often reflected in their emotional responses to academic situations. This approach can spark positive academic emotions, thereby enhancing the overall psychological well-being of students [31]. Schools should promote the construction of harmonious teacher–student relationships, discover and intervene in college students’ mental health problems in a timely manner, help students solve problems and difficulties encountered in learning life, enhance emotional exchanges between teachers and students, actively care for and love students, respect students’ personalities, and improve student happiness.

(4) Carry out a variety of mental health education activities to enhance students’ confidence and interest in learning. 

Colleges should carry out a variety of mental health education activities. For example, theme class meetings like “how to deal with negative emotions”, “small happiness in life”, “gratitude education”, etc. should be held regularly to cultivate students’ positive thinking. Moreover, college should fully play the main channel role of classroom teaching in cultivating students’ psychological quality and carry out extracurricular activities to enhance students’ confidence and interest in learning through various channels so that students can experience happiness in activities and establish correct outlooks on life and values [53].

## 5. Conclusions

Despite the increasing recognition and empirical support for the dual-factor model of mental health, which elucidates the interplay between well-being and psychopathology, the specific profiles associated with dual-factor mental health and their predictors are still unclear, especially among Chinese university students. 

The outcomes of this study demonstrated that, in comparison to the single-factor structural model, the dual-factor structural model exhibited a superior fit across various goodness-of-fit indices. The analysis revealed the emergence of three distinct groups, namely, Flourishing, Vulnerable, and Troubled. These findings enhance the comprehension of the heterogeneous patterns of dual-factor mental health during the university years and advocate against adopting a universal intervention approach. Instead, they underscore the necessity of universally screening students for dual-factor mental health to design more nuanced intervention initiatives tailored to the unique attributes of specific groups. Furthermore, the identification of these three distinct groups, as opposed to a four-group classification system, suggests the importance of considering cultural and methodological factors in future research [29]. This is crucial as mental health profiles might not seamlessly translate across different cultural contexts. The discernment of specific academic emotional factors linked to various mental health statuses provides valuable insights into understanding the underlying sources of group disparities. This understanding should guide educational institutions in devising assessment and intervention strategies that address psychological concerns and facilitate the promotion of optimal mental well-being.

## Figures and Tables

**Figure 1 healthcare-11-02719-f001:**
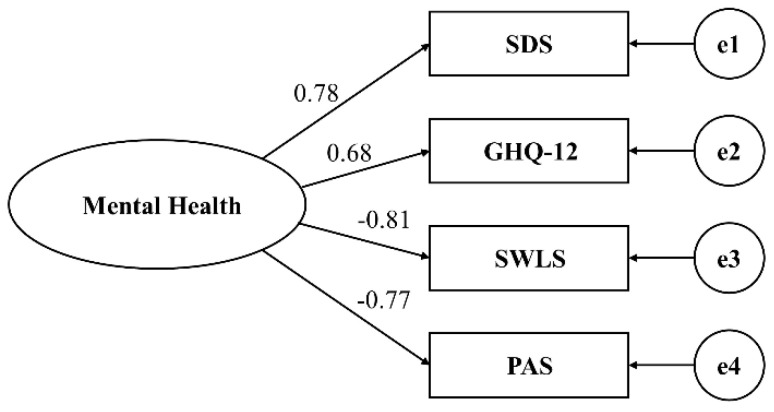
Mental health single-factor model (Model 1).

**Figure 2 healthcare-11-02719-f002:**
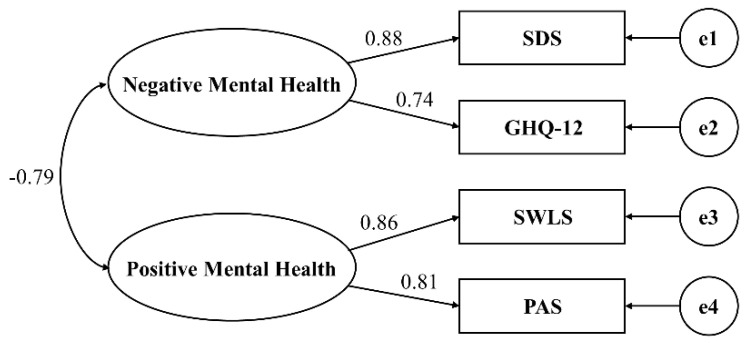
Dual-factor model of mental health (Model 2).

**Figure 3 healthcare-11-02719-f003:**
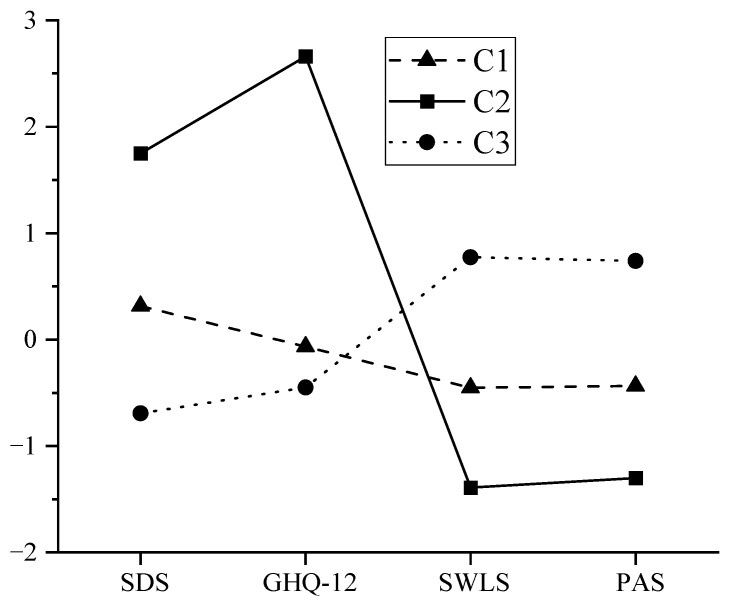
Latent profile plot of university students’ mental health.

**Table 1 healthcare-11-02719-t001:** Sociodemographic distribution of study participants (N = 2277).

Demographic	Items	Frequency	Percentage (%)
Gender	Male	1402	61.6
	Female	875	38.4
Birth Place	Rural	847	37.2
	Urban	1430	62.8
Major	Humanities	293	12.9
	Science	1598	70.2
	Arts	5	2.2
	Medical	336	14.8

**Table 2 healthcare-11-02719-t002:** Model fit of different mental health models.

Fit Indices	Chi-Square to Degrees of Freedom (χ^2^/df)	Comparative Fit Index (CFI)	Normed Fit Index (NFI)	Root Mean Square Error of Approximation (RMSEA)	Standardized Root Mean Square Residual (SRMR)
Model 1	153.029	0.925	0.924	0.257	0.052
Model 2	4.074	0.999	0.999	0.037	0.005

**Table 3 healthcare-11-02719-t003:** Fit statistics, classification indices, and class sizes for each model.

	K	LL	AIC	BIC	aBIC	Entropy	LMR(p)	BLRT(p)	Class Probabilities
C = 1	8	−13,042.883	26,101.766	26,147.684	26,122.267	--	--	--	--
C = 2	13	−11,535.812	23,097.624	23,172.241	23,130.938	0.960	<0.001	<0.001	0.889, 0.111
C = 3	18	−10,827.592	21,691.184	21,794.500	21,737.311	0.809	<0.001	<0.001	0.476, 0.441, 0.083
C = 4	23	−10,389.900	20,825.800	20,957.815	20,884.740	0.831	0.008	<0.001	0.492, 0.116, 0.348, 0.044
C = 5	28	−10,204.524	20,465.048	20,625.762	20,536.801	0.848	0.006	<0.001	0.471, 0.044, 0.128, 0.338, 0.019

**Table 4 healthcare-11-02719-t004:** Average membership probabilities (columns) for each latent class (rows).

	C1	C2	C3
C1	0.906	0.087	0.008
C2	0.093	0.907	0.000
C3	0.048	0.000	0.952

**Table 5 healthcare-11-02719-t005:** Multinomial logistic regression of academic emotions on profiles.

Model	Predictor	b	SE	χ^2^	P	OR	LLCI	ULCI
Flourishing vs. Vulnerable		−0.052	0.009	−5.996	<0.001	0.949	0.933	0.965
	PHA *	0.096	0.014	6.673	<0.001	1.101	1.071	1.133
	NLA *	0.006	0.009	0.644	0.519	1.006	0.988	1.024
	PLA	0.096	0.014	7.065	<0.001	1.101	1.072	1.131
Troubled vs. Vulnerable	NHA *	0.079	0.014	5.474	<0.001	1.082	1.052	1.114
	PHA	−0.087	0.021	−4.138	<0.001	0.917	0.88	0.955
	NLA *	0.012	0.014	0.863	0.388	1.012	0.984	1.041
	PLA	−0.081	0.017	−4.878	<0.001	0.922	0.893	0.953

* PHA stands for positive high-arousal; PLA stands for positive low-arousal; NHA stands for negative high-arousal; and NLA stands for negative low-arousal.

## Data Availability

Data are available upon request to the first author.

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
