# Peer review of "Latent Profile Analysis of Mental Health among Chinese University Students: Evidence for the Dual-Factor Model"

_healthcare, 2023, doi:10.3390/healthcare11202719_

Round 1

Reviewer 1 Report

Upon reading the manuscript, I found it might be more suitable for a psychological journal than Healthcare.

1.       The introduction is engaging and effectively contextualizes the research. However, there is an inconsistency in the terminology used. In the abstract, the authors mention three distinctive groups of people under the dual-factor model of mental health, but in the introduction, they identify a fourth group as "symptomatic but content." It's crucial for the authors to maintain consistency in their nomenclature.

2.       The background section, explaining the methodology, is well-detailed and provides intriguing insights.

3.       Regarding the demographic distribution, I suggest adding information about the age and stage of university students. Age and stage can significantly influence the development of coping mechanisms and, consequently, students' categorization in terms of mental health.

4.       Reading the results section can be challenging. The authors make the results overly complex without adequately explaining the specific analyses conducted. While this may not be an issue for specialized readers, it poses a significant challenge for the average reader seeking clarity in the results.

-The presentation in Table 3 lacks a clear explanation, rendering it unreadable.

- In Table 4, three groups suddenly emerge without any explanation of how this came about. While this issue is addressed later in the discussion, it raises questions about its significance.

-Figure 3 displays a latent profile plot, but its significance is unclear due to a lack of explanation. It appears as if the data are presented without sufficient accompanying context and explanation.

-Table 5 is more comprehensible thanks to the explanatory paragraph preceding it. However, overall, the results section of the paper is quite brief and lacks the necessary clarity for an average reader.

Providing links to the relevant questions that were asked and the calculations of scores would enhance understanding.

5.       The discussion section does not address the age of students, which is a crucial indicator of coping mechanisms in mental health. Therefore, it should be appropriately discussed or analysed to provide a more comprehensive perspective.

In summary, while the statistical aspects and the introduction of the paper are well-executed, there is a need for better linkage between the results and their practical implications. The brief paragraph in section 4.3 that discusses the significance of the findings does not effectively convey how useful this analysis might be for universities as institutions.

Top of Form

Author Response

Dear reviewer,

Thank you for providing many helpful comments to improve the paper. We appreciate your time and effort in this regard. Please see the attachment.

Best Regards~!

Reviewer 2 Report

 1.      The research questions or aims should be clarified point by point in section 1.4.

2.      In section 2.2.6, you said you used Amos 21. In line 320, you said you used Amos 24. Please explain these inconsistent statements.

3.      The process of LPA was not clearly presented. The authors should address their approaches of identifying profiles step by step with clear subheading in the paragraphs. Also, this process should be described in detail in the Methods section.  

4.      Discussion about the practical and clinical implications of this study were insufficient. The section 4.4 should be boosted with more discussions on practical and clinical implications, especially in your study context (Chinese college students).

5.      Limitations and future studies should be added in a separate section in Discussion.

6.      The conclusion is too long. 

None

Author Response

Dear reviewer,

Thank you for providing many helpful comments to improve the paper. We appreciate your time and effort in this regard.

Best Regards~!

Reviewer 3 Report

 I found your MS well organized and cleanly argued. I think it fills a niche in the literature quite nicely. I believe the methodology well-wrought and the findings generally well presented. I do believe the article would benefit from a thorough copy editing.  All of this said, I have three questions. First, I do not believe the argument advanced in line 387-388 is substantiated in any way and is not material to the general argument in any case and so I suggest it be eliminated. Second, there is a glitch in lines 397-398 and it therefore makes no sense as written?  I believe you mean a priori logic? Finally, I would like to see more evidence of the “externalizing measures” referenced in lines 459-460. Such would help readers understand more completely the distinction you are seeking to press here. This distinction is important to explaining your study results. Overall, I found your effort fascinating and well done.

I think this piece would benefit from a thorough copy editing as I note above.

Author Response

(The authors gave the same response as above.)

Round 2

Reviewer 1 Report

The authors responded to my comments in a very comprehensive manner and provided extensive review and corrections of their manuscript.